# Peptide transporter structure reveals binding and action mechanism of a potent PEPT1 and PEPT2 inhibitor

Mirko Stauffer [1,3], Jean-Marc Jeckelmann[1,3], Hüseyin Ilgü[1], Zöhre Ucurum[1], Rajendra Boggavarapu[1,2] & Dimitrios Fotiadis [1✉]

Inhibitors for membrane transporters have been shown to be indispensable as drugs and tool compounds. The proton-dependent oligopeptide transporters PEPT1 and PEPT2 from the SLC15 family play important roles in human and mammalian physiology. With Lys[Z(NO$_2$)]-Val (LZNV), a modified Lys-Val dipeptide, a potent transport inhibitor for PEPT1 and PEPT2 is available. Here we present the crystal structure of the peptide transporter YePEPT in complex with LZNV. The structure revealed the molecular interactions for inhibitor binding and a previously undescribed mostly hydrophobic pocket, the PZ pocket, involved in interaction with LZNV. Comparison with a here determined ligand-free structure of the transporter unveiled that the initially absent PZ pocket emerges through conformational changes upon inhibitor binding. The provided biochemical and structural information constitutes an important framework for the mechanistic understanding of inhibitor binding and action in proton-dependent oligopeptide transporters.

[1] Institute of Biochemistry and Molecular Medicine, and Swiss National Centre of Competence in Research (NCCR) TransCure, University of Bern, Bern, Switzerland. [2] Present address: Department of Physiology and Biophysics, Case Western Reserve University, Cleveland, OH, USA. [3] These authors contributed equally: Mirko Stauffer, Jean-Marc Jeckelmann. ✉email: dimitrios.fotiadis@ibmm.unibe.ch

The proton-dependent oligopeptide transporter (POT) family[1–4] (also referred to as the peptide transporter (PTR) family[2,5]) is a group of secondary active symporters[6,7], which belong to the major facilitator superfamily (MFS)[8,9]. They actively transport and accumulate their substrates across cellular membranes using the proton electrochemical gradient[3]. Valuable information on the transport function of POTs was obtained from bacterial family members, e.g., from the well-studied peptide transporters from the bacterium *Escherichia coli*[10–17]. In humans, the POTs PEPT1 (SLC15A1) and PEPT2 (SLC15A2) have important physiological and pharmacological roles[18,19]. While PEPT1, a high-capacity, low-affinity transporter mainly expressed in the intestine, takes up dietary di- and tripeptides, PEPT2, a low-capacity, high-affinity transporter is predominantly expressed in the kidney, where it is responsible for the reabsorption of di- and tripeptides[18]. Both transporters are also expressed in other human and mammalian cell types[18]. PEPT1 and PEPT2 have pharmacological relevance, as they were shown to translocate a wide range of peptidomimetic drugs and prodrugs[20] such as antibiotics[21–24] (e.g., cefadroxil and cefprozil), antivirals[25] (e.g., valaciclovir), protease inhibitors[26] (e.g., bestatin) or Parkinson medications[25] (e.g., L-dopa-L-Phe, L-dopa). Recently, inhibition of PEPT1 in inflammatory bowel disease patients was proposed as therapeutic approach to block membrane transport of toxic bacterial products[27].

Over the last years, several crystal structures of bacterial homologues were solved in the apo form[28–36] and with bound dipeptides[37–39], tripeptides[37,38,40] and other peptidomimetic ligands[30,36,41–43]. Although variations in the binding mechanism of different ligands were observed[39,40,43], several conserved binding pocket residues were identified, which interact with the peptide/peptidomimetic backbone of most ligands[44]. Furthermore, hydrophobic pockets accommodating the N-terminal (P1) and C-terminal (P2) amino acid side chains of dipeptides[38,39], as well as one accommodating the additional side chain in tripeptides[40] (P3), were described. It was proposed that the multispecificity observed in POTs arises in part due to water network modulations and movements of (predominantly aromatic) residues in those pockets[39]. As other MFS transporters, POTs translocate their substrates using an alternate access mechanism, involving outward-open and outward-facing occluded as well as inward-open and inward-facing occluded states[6,7,9]. Until recently, all structures of POTs were either in the inward-open or inward-facing occluded conformation[3]. Cryo-electron microscopy studies provided now also outward-open and outward-facing occluded states with the structures of mammalian PEPT1[45] and PEPT2[45,46].

Thorough investigation of transport processes by membrane proteins is facilitated by the availability of specific high-affinity inhibitors. A range of modified dipeptides that specifically inhibit mammalian peptide transporters have been developed[47–51]. Especially, dipeptides with a chemically modified amino acid side chain group containing lysine as N-terminal residue were shown to inhibit the mammalian peptide transporters PEPT1 and PEPT2 with high affinity, e.g., Lys[Z(NO$_2$)]-Val (LZNV; $K_i = 2$ μM[51–53]). Although extensive structure-activity relationship studies allowed the identification of structural features important for increasing affinity[50,52], no structural evidence for the exact molecular interactions between those inhibitors and peptide transporters could be obtained so far.

We determined previously the structure of the wild-type POT family member YePEPT from the bacterium *Yersinia enterocolitica*[33]. To understand the molecular interactions of the potent PEPT1 and PEPT2 inhibitor LZNV, we solved here the crystal structure of YePEPT-K314A (YePEPT$^{K314A}$) in the apo form and in complex with LZNV. The co-crystal structure allowed for the elucidation of the molecular interactions between a POT and the inhibitor LZNV. In addition, a new mostly hydrophobic pocket accommodating the 4-nitro-benzyloxycarbonyl moiety (Z(NO$_2$)) of LZNV was identified and described. Mutagenesis of single amino acid residues in the binding pocket was used to evaluate protein-ligand interactions observed in the structure. Furthermore, comparison of the apo structure with the LZNV bound structure of YePEPT$^{K314A}$ enabled the identification of conformational changes of amino acid side chains and a rigid body movement of the N-terminal bundle upon binding of the inhibitor. The observed conformational changes correspond to the transition from the inward-open towards an inward-facing partially occluded conformation upon inhibitor binding. Finally, and based on obtained and available information, we propose a general mechanism of action for the inhibition of POTs by LZNV.

## Results and discussion

**Thermostability analysis of wild-type YePEPT and YePEPT$^{K314A}$ in the absence and presence of LZNV.** Ligand stabilisation of detergent purified wild-type YePEPT (YePEPT$^{WT}$) by different concentrations of LZNV was analysed using a label-free differential scanning fluorometry-based thermal shift assay (TSA) (Fig. 1a). Increasing concentrations of LZNV did not significantly change the inflection temperature ($T_i$) of YePEPT$^{WT}$ indicating that LZNV is not stabilising YePEPT$^{WT}$ through protein-ligand interactions. This might be due to steric hindrance by amino acid side chains in the substrate-binding pocket. In the previously described mutant YePEPT$^{K314A}$, the long and charged side chain K314 was replaced with alanine to increase the volume of the binding pocket and therefore allow binding of larger substrates to YePEPT[33]. When repeating the TSA experiment with the YePEPT$^{K314A}$ variant, the $T_i$ increased significantly in a LZNV concentration-dependent manner (Fig. 1a), indicating LZNV binding and stabilisation of YePEPT$^{K314A}$.

**Substrate transport inhibition by LZNV using *E. coli* cells overexpressing YePEPT$^{K314A}$.** It was evaluated if LZNV inhibits transport in an uptake assay using *E. coli* cells overexpressing YePEPT$^{K314A}$ and the radioligand [$^3$H]Ala-Ala as reporter substrate. In a first step, the Michaelis-Menten constant ($K_m$) of YePEPT$^{K314A}$ for the substrate Ala-Ala was determined to be 57.3 μM (Fig. 1b). This $K_m$ is about four times lower than that of YePEPT$^{WT}$ (~200 μM[33]) indicating a higher affinity of YePEPT$^{K314A}$ for the substrate Ala-Ala. To examine the effect of LZNV on Ala-Ala transport by YePEPT$^{K314A}$, we determined a dose-response curve of LZNV for Ala-Ala transport, which yielded an $IC_{50}$ of 7.6 μM (Fig. 1c). Considering this $IC_{50}$, LZNV inhibition of Ala-Ala uptake in YePEPT$^{K314A}$ is significantly better than with the peptide transporter DtpA from *E. coli* ($K_i = 43$ μM)[54].

The presented TSA and transport inhibition experiments led to the conclusion that LZNV binds to detergent-solubilized YePEPT$^{K314A}$ and inhibits Ala-Ala uptake in a cell-based uptake assay. Importantly, this opened the possibility to explore the molecular interactions of a POT with the potent mammalian PEPT1 and PEPT2 inhibitor by co-crystallisation of YePEPT$^{K314A}$ with LZNV.

**Overall structure of YePEPT$^{K314A}$ with LZNV.** Three-dimensional (3D) crystals of YePEPT$^{K314A}$ in the presence of LZNV were successfully grown and analysed by X-ray diffraction. The crystal structure of YePEPT$^{K314A}$ was solved to 2.66 Å resolution by molecular replacement using the coordinates of YePEPT$^{WT}$ (PDB ID code: 4W6V[33]) (Supplementary Table 1). The quality of the obtained electron density map is illustrated in

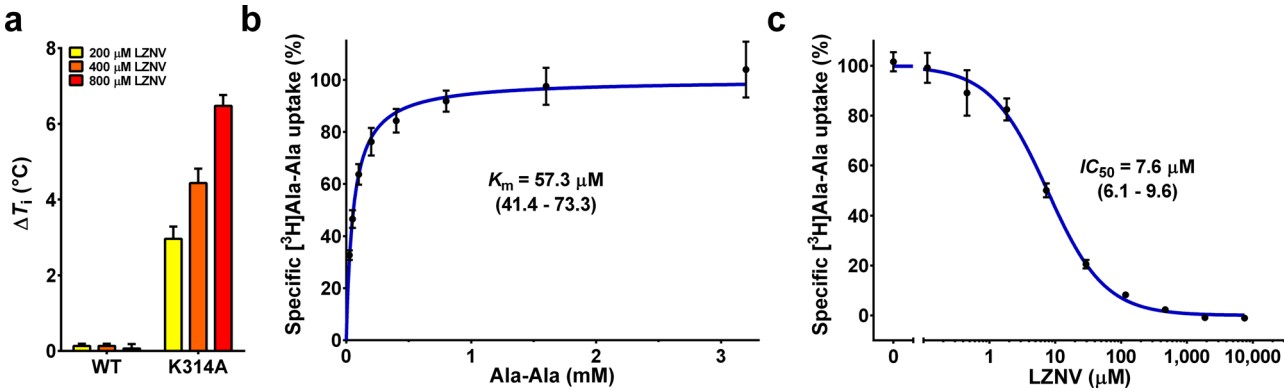

**Fig. 1 Influence of LZNV on the thermostability and function of YePEPT. a** Thermal shift assay using YePEPT$^{WT}$ and YePEPT$^{K314A}$, and LZNV. The $T_i$-values of YePEPT$^{WT}$ and YePEPT$^{K314A}$ were determined at different concentrations of LZNV (200 μM, 400 μM and 800 μM). Δ$T_i$-values at specified LZNV concentrations were calculated by subtraction of the $T_i$-value in the absence of LZNV. Data points represent mean ± SD of at least three independent experiments. **b** Kinetics of [³H]Ala-Ala uptake in *E. coli* cells transformed with the YePEPT$^{K314A}$ construct. Data points represent uptake of [³H]Ala-Ala normalised to $V_{max}$ ± SEM of three independent experiments, each at least in triplicate. **c** $IC_{50}$ determination of YePEPT$^{K314A}$ for LZNV by heterologous competition. Data points represent uptake of [³H]Ala-Ala normalised to the uninhibited signal ± SEM of three independent experiments, each at least in triplicates. Numbers in brackets below $K_m$- and $IC_{50}$-values represent 95% confidence intervals. If not visible, error bars are smaller than symbols.

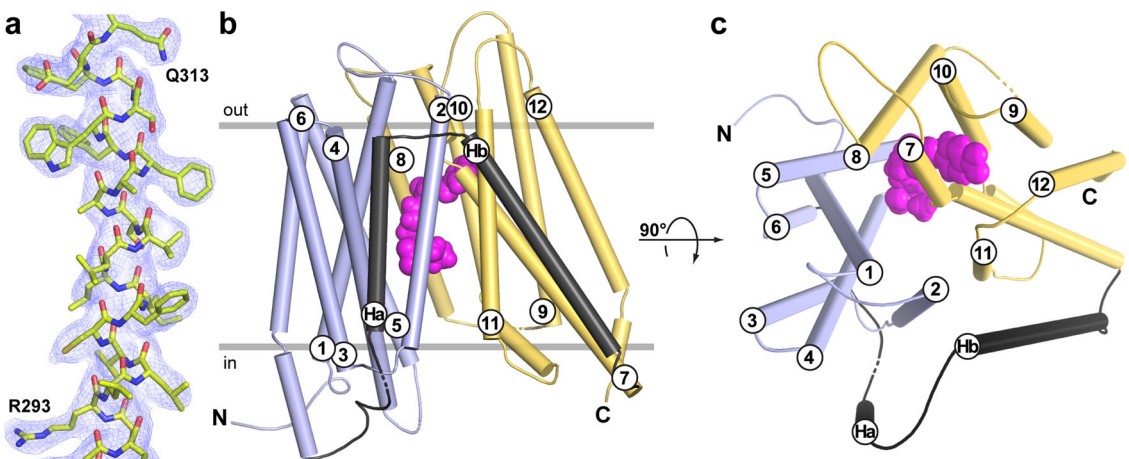

**Fig. 2 Electron density and overall structure of YePEPT$^{K314A}$ with bound LZNV. a** Electron density map of the YePEPT$^{K314A}$ crystal structure. Displayed is the 2Fo-Fc electron density map of a structural fragment of TM7 contoured at 1.0 σ and coloured in blue, and the YePEPT$^{K314A}$ structure as yellow sticks. The shown fragment of TM7 corresponds to the sequence 293-**R**LLVCFILLVSAAFFWSAFE**Q**-313, whereby residues in bold are labelled in the panel. **b** and **c** Overall structure of YePEPT$^{K314A}$ with bound LZNV in an inward-facing conformation. The structure is viewed as from the membrane plane (**b**) and from the periplasm (**c**). Grey horizontal lines in **b** indicate the orientation of YePEPT$^{K314A}$ in the biological membrane according to the PPM-server[70]. In **b** and **c**, the bound LZNV molecule is shown in purple as space-filling model. The N- and C-termini are indicated in capital letters and all 14 TMs are labelled N-terminally (**b**) or at the periplasmic side (**c**). The pseudo-twofold symmetrical N- and C-terminal six-helix bundles are coloured in blue and yellow, respectively, whereas the two bundles connecting helices (Ha and Hb) are depicted in black.

Fig. 2a. The structure contains 14 transmembrane helices (TMs) with an N- and a C-terminal six-helix bundle (Fig. 2b, c; bundles in blue and yellow) formed by TM1-6 and TM7-12, respectively. These bundles form the canonical major facilitator superfamily (MFS) fold[55,56] with a pseudo-twofold symmetry relation to each other. The two terminal bundles are connected by the two additional TMs Ha and Hb (Fig. 2b, c; depicted in black) in a hairpin-like structure, which is typical for prokaryotic peptide transporters[44,57]. The structure of YePEPT$^{K314A}$ is in an inward-facing conformation, with a central, conical cavity facing the cytosol. Extra density in the conical cavity (Supplementary Fig. 1) was attributed to one LZNV molecule (Fig. 2b, c; depicted in purple). The location of the LZNV molecule corresponds to the substrate-binding pocket according to previously published POT crystal structures in complex with ligands[30,36–43].

**Protein interactions with the dipeptide backbone of LZNV.** One molecule of the inhibitor LZNV was found bound to YePEPT$^{K314A}$ (Fig. 2b, c; see also Supplementary Fig. 1 for polder OMIT map[58]). The LZNV molecule is an analogue of the Lys-Val dipeptide, with a 4-nitro-benzyloxycarbonyl (Z(NO₂)) protecting group attached to the amino group of the lysine residue side chain. It contains five functional groups, i.e., a nitro-, a carbamate-, a primary amino-, an amide- and a carboxyl group (Fig. 3a and Supplementary Fig. 2a), potentially involved in polar or ionic interactions with the protein. All polar and ionic interactions within a distance of ≤3.5 Å from amino acid residues to the LZNV dipeptide backbone were investigated in more detail (Fig. 3b, c and Supplementary Fig. 3). The primary amino group is involved in an ionic interaction with the carboxyl group of E420 (TM10) (Fig. 3c). The corresponding conserved residue in

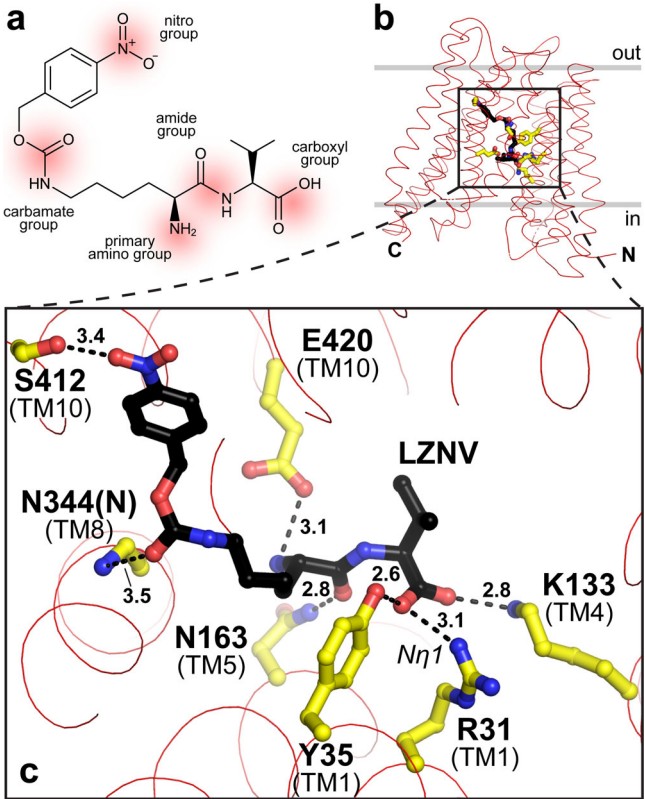

**Fig. 3 LZNV binding pocket in YePEPT$^{K314A}$ and molecular protein-ligand interactions. a** Chemical structure of LZNV. Polar and ionic groups are labelled and highlighted in red. **b** Overall structure of YePEPT$^{K314A}$ with bound LZNV oriented in the lipid membrane according to the PPM-server[70]. The location of the binding pocket in the transporter is indicated by a black box. **c** Detailed view into the LZNV binding pocket. Polar and ionic interactions of LZNV (black sticks) with amino acid residues (yellow sticks) located in the binding pocket of YePEPT$^{K314A}$ are displayed. All polar and ionic interactions within a distance ≤3.5 Å from amino acid residues to the ligand are shown as dashed lines (distances in Å). Amino acids are labelled in the one-letter code and corresponding TMs are indicated in brackets. For N344, the interaction with the main-chain nitrogen is indicated by N344(N).

other POTs was previously shown to interact with the primary amino group of bound dipeptides[37–39]. The spatial orientation of the carboxyl group of E420 (TM10) appears to be guided by hydrogen bonds (H-bonds) with the side chain oxygen atoms of N344 (TM8) and Y159 (TM5) (Supplementary Fig. 2b, d). The carbonyl oxygen atom of the LZNV amide group accepts an H-bond from the amide nitrogen atom of N163 (TM5) (Fig. 3c). The carboxyl group of LZNV interacts with the Nη1 of the guanidinium group of R31 (TM1), with the hydroxyl group of Y35 (TM1), and with the side chain amine of K133 (TM4) (Fig. 3c). Interactions of the corresponding conserved residue side chains in other POTs with functional groups of dipeptide backbones were previously described[37–39] (Supplementary Fig. 4). For LZNV binding, the carboxyl group of E30 (TM1) appears to orientate the guanidinium group of R31 (TM1) in space through interactions between the oxygen atom Oε1 of E30 (TM1) and Nη2 of R31, as well as between Oε2 of E30 and Nε of R31 (Supplementary Fig. 2c, d).

**Protein interactions with the side chains of LZNV.** The valine side chain of LZNV points towards the previously described

Pocket 2[38] (P2, Fig. 4a, in green) consisting of Y73, W308, E312, S451 (corresponding conserved residues in PepT$_{St}$: Y68, W296, E300 and S431). The Cβ to Cε-atoms of the modified lysine side chain of LZNV are located in the previously described hydrophobic Pocket 1[39] (P1, Fig. 4a, in purple) consisting of Y35, S166, N344 and A345 (corresponding residues in PepT$_{St}$: Y30, A159, N328 and P329). This pocket was shown to accommodate the side chain of the N-terminal amino acid of di- and tripeptide ligands in structures of other peptide transporters[38–40]. Due to the additional Z(NO$_2$) moiety, the whole modified Lys side chain is too long and bulky to fit into P1 alone. Thus, it continues into another previously undescribed mostly hydrophobic pocket (PZ) comprised of 9 residues within a distance of 4.0 Å to the Z(NO$_2$) moiety of LZNV, i.e., residues F311, Q313, A314, F318, F386, M389, S412, I413 and L416 (Fig. 4a, in yellow). Most of the residues forming the PZ pocket are comparable, some even identical in bacterial peptide transporters, as well as in PEPT1 and PEPT2 (Supplementary Table 2), we therefore expect a comparable LZNV binding mode in those transporters. Interestingly, the access to PZ was generated by the K314A mutation, since K314 in the structure of YePEPT$^{WT}$ (PDB ID code: 4W6V) is blocking the entry to PZ (Fig. 4b, in red)[33]. This explains why LZNV did not stabilise YePEPT$^{WT}$ (Fig. 1a) and confirms our hypothesis that the removal of the large K314 side chain generates the space needed for the binding of large ligands such as LZNV.

The hydrophobic Z(NO$_2$) moiety of LZNV is accommodated in the PZ pocket close to hydrophobic amino acid side chains (Fig. 4). Analysis using the protein-ligand interaction profiler (PLIP[59]) revealed that F318 forms a π-stacking interaction with the 4-nitrophenyl group of LZNV (Fig. 4b, cyan dashed line). In addition, two polar interactions within a distance of ≤3.5 Å from amino acid residues to LZNV were observed (Fig. 3c). The carbonyl oxygen of the carbamate group accepts a H-bond from the backbone nitrogen atom of N344 (N344(N), TM8). The observed protein backbone interaction might explain why modified dipeptides were demonstrated to have an enhanced affinity for PEPT1 when the modification on the lysine side chain is attached via a carbamate group linker[52]. The nitro group forms a weak H-bond with the hydroxyl group of S412 (TM10) in the PZ pocket (Fig. 3c). In summary, within a distance of ≤3.5 Å seven polar or ionic protein-ligand interactions were identified of which six interactions are with protein amino acid side chain residues and one interaction is with the protein backbone, i.e., with N344(N). Additionally, the 4-nitrophenyl group is involved in hydrophobic interactions in the PZ pocket and forms a π-stacking interaction with F318.

**Investigation of protein-LZNV interactions by mutagenesis.** To verify the six observed, polar or ionic protein side chain to ligand interactions and to estimate their contribution to LZNV binding, the corresponding amino acids were each separately mutated to alanine. Purified proteins of corresponding YePEPT mutants were then probed by TSA using varying LZNV ligand concentrations (Fig. 5). YePEPT$^{K314A-S412A}$ still showed a strong dose-dependent stabilisation by LZNV, indicating that the interaction of the nitro group with S412 is weak and not of high importance for ligand binding. In contrast, the interaction of the LZNV primary amino group with E420 seems to be crucial for ligand binding, since for the YePEPT$^{K314A-E420A}$ variant no ligand-dependent stabilisation could be observed. Similarly, for the YePEPT$^{K314A-N163A}$ variant, removal of the H-bond between N163 and the carbonyl oxygen atom of the LZNV amide group leads to loss of stabilisation. When the three residues interacting with the carboxyl group of LZNV (i.e., R31, Y35 and K133) were

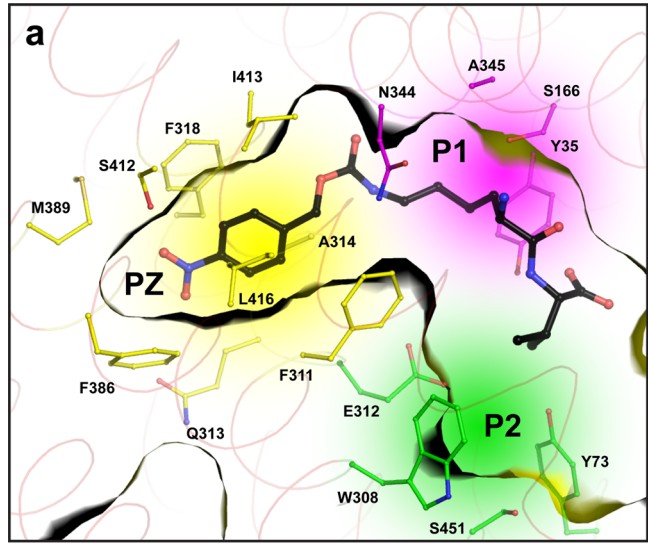

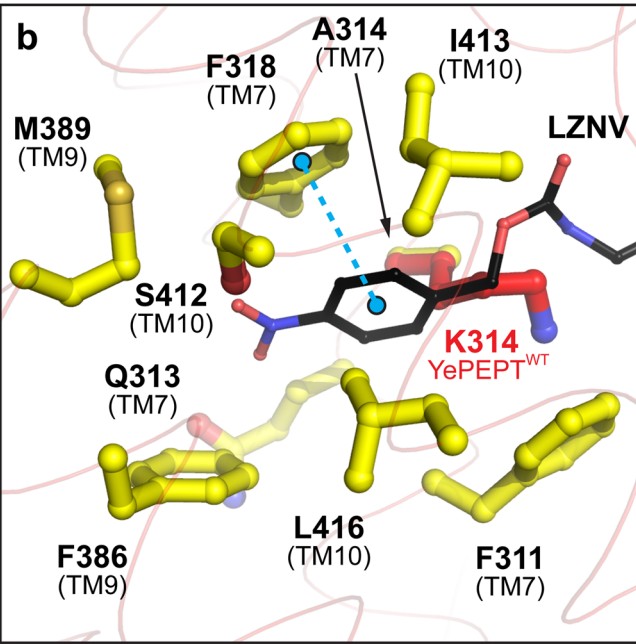

**Fig. 4 LZNV binding pockets of YePEPT$^{K314A}$. a** Accommodation of substrate/ligand side chains in the previously described pockets P1 (purple) and P2 (green), and of the Z(NO$_2$) moiety in the newly identified pocket PZ (yellow). **b** Detailed view into the PZ pocket, accommodating the Z(NO$_2$) moiety of LZNV. The π-stacking interaction between F318 and the 4-nitrophenyl group of LZNV is displayed with two balls located in the centre of the aromatic rings and connected by a dashed line (cyan). The position of K314 in the structure of YePEPT$^{WT}$ is shown in red. Protein side chains involved in pocket formation are depicted as thin (**a**) or thicker (**b**) sticks and colour-coded according to corresponding pocket colour. Amino acids are labelled in the one-letter code and corresponding TMs are indicated in brackets.

each independently mutated to alanine, ligand-dependent protein stabilisation was abolished for all resulting double mutants (i.e., YePEPT$^{K314A-R31A}$, YePEPT$^{K314A-Y35A}$ and YePEPT$^{K314A-K133A}$). Therefore, protein interactions with the carboxyl group of the LZNV seem to be crucial for successful ligand binding.

**Conformational changes of YePEPT$^{K314A}$ upon LZNV binding.** To investigate potential structural changes upon binding of LZNV, the structure of YePEPT$^{K314A}$ was also solved in the

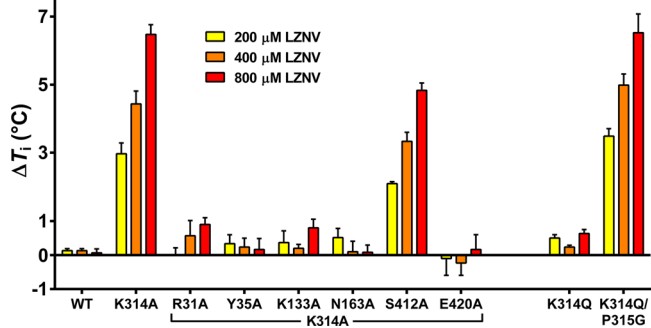

**Fig. 5 Thermal stabilisation of different YePEPT variants by LZNV.** The $T_i$-values of different YePEPT variants were determined at three different concentrations of LZNV, i.e., 200 μM, 400 μM and 800 μM. Δ$T_i$-values at the specified LZNV concentrations were determined by subtraction of the $T_i$-value in the absence of LZNV. Data points represent mean ± SD of at least three independent experiments.

apo state to a resolution of 2.93 Å (Supplementary Table 1). A superposition of both structures (Fig. 6a) showed that upon LZNV binding, the N-terminal six-helix bundle undergoes a rigid body movement of about 5° towards the C-terminal bundle with the hinge located towards the periplasmic region of the N-terminal bundle. Similar domain rotations could be observed by comparison of inward-open and inward-facing occluded structures of PepT$_{St}$[34] and MFS transporters in general[9]. Furthermore, the structural alignment allowed the identification of three aromatic amino acid side chains with significantly different rotamer conformations between the two structures (Fig. 6b). F318 adopts another rotamer conformation upon LZNV binding, generating the necessary space for the Z(NO$_2$) moiety of LZNV to bind deep into the PZ site (Fig. 6b). Additionally, in this rotamer conformation, F318 is oriented correctly to form the observed π-stacking interaction with the 4-nitrophenyl group (Fig. 4b). Upon LZNV binding, the hydroxyl group of Y35 (TM1) moves by ~2 Å compared to the apo structure. This movement is beneficial for the establishment of the observed interaction of the Y35 (TM1) hydroxyl group with the carboxyl group of LZNV (Fig. 3c). Further, the side chain of Y159 (TM5) adopts another rotamer conformation upon LZNV binding, which is characterised by a significant movement of 9.5 Å of the hydroxyl group. As Y159 (TM5) is located in the region with the largest displacement due to the rigid body movement, the Cα-atom of Y159 shows a larger shift of 1.5 Å compared to the other two residues, i.e., Y35 (0.4 Å) and F318 (0.3 Å). As mentioned previously, Y159 is involved in the spatial orientation of the carboxyl group of E420 for LZNV binding indicating an indirect contribution in binding of the ligand (Supplementary Fig. 2b). Interestingly, in PepT$_{So}$ the corresponding conserved residue (Y154) was demonstrated to be part of the intracellular gate in an inward-facing occluded conformation[28]. However, the amino acid side chains corresponding to other residues involved in forming the gate of PepT$_{So}$, did not undergo significant conformational changes upon LZNV binding in YePEPT$^{K314A}$. Therefore, the alternative rotamer conformation of Y159 in the LZNV bound structure of YePEPT$^{K314A}$ might be the first step in the formation of the intracellular gate. Another feature in the transition to the inward-facing occluded state in peptide transporters is the bending of the cytoplasmic ends of certain transmembrane helices, in particular TM10 and TM11[34,39]. No such movement could be observed in YePEPT$^{K314A}$ upon LZNV binding. Importantly, comparison of the apo (Fig. 6c) with the LZNV bound structure (Fig. 6d) unveiled that the PZ

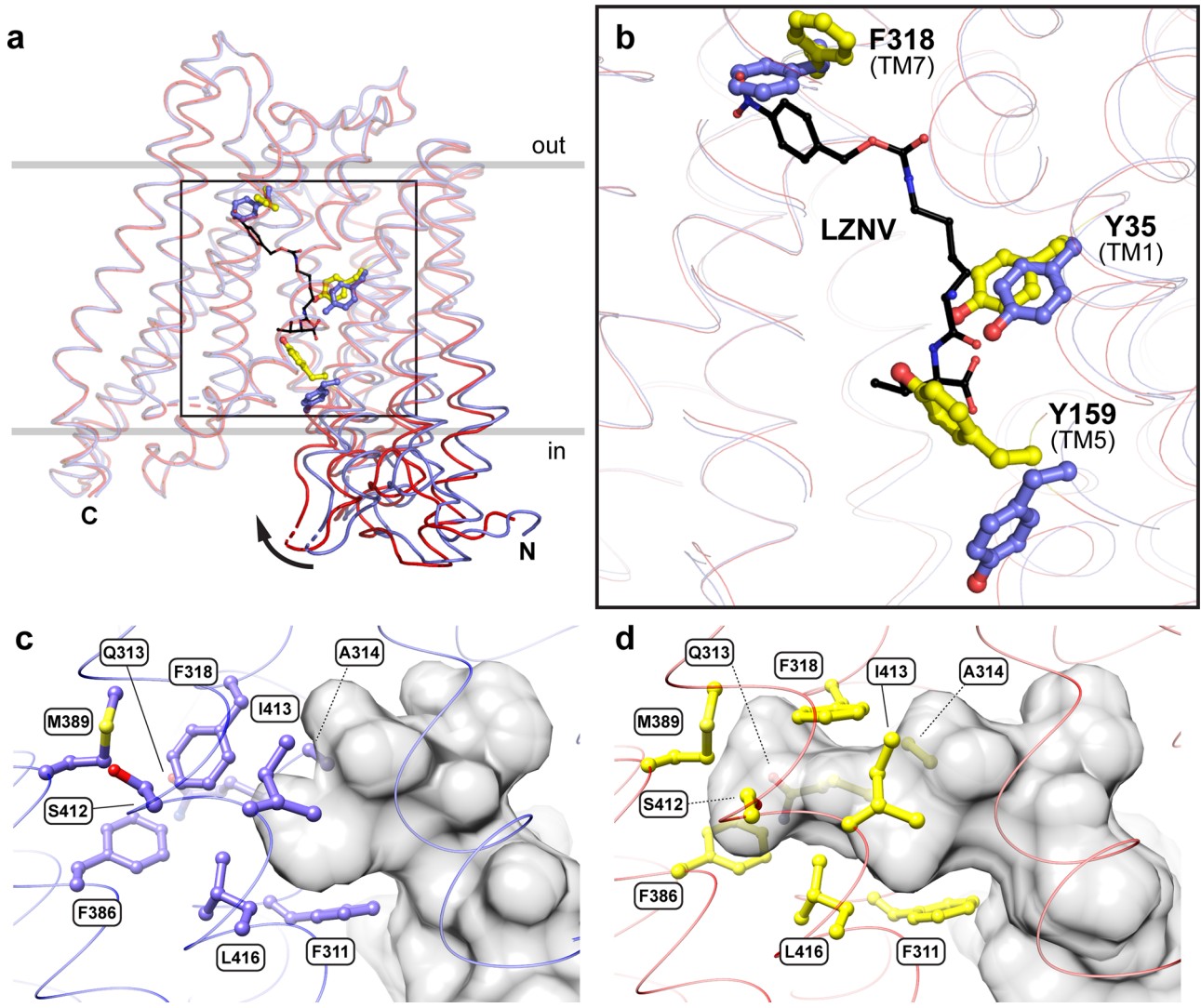

**Fig. 6 Conformational changes of YePEPT^K314A and formation of the PZ pocket upon binding of LZNV. a** Structural alignment of overall structures of the apo (blue ribbon) and the LZNV bound YePEPT^K314A (red ribbon). The location of largest displacement between the apo and LZNV bound structures is found in the N-terminal six-helix bundle facing the cytosol (in), as indicated by an arrow. **b** The LZNV binding site, marked in (**a**) by a black box, is displayed enlarged. Side chains undergoing significant conformational changes upon LZNV binding are depicted as sticks and coloured in blue (apo structure) and yellow (LZNV bound structure). The LZNV molecule is displayed in a smaller stick-diameter mode compared to the side chain moieties and coloured in black. Amino acids are labelled in the one-letter code and corresponding TMs are indicated in brackets. View into the PZ pocket location in the apo (**c**) and LZNV bound structures (**d**). Side chains from residues forming the PZ pocket are labelled and coloured as in **b**. Solvent accessible volume is displayed and coloured in grey (**c**, **d**).

pocket is initially absent and emerges through conformational changes upon inhibitor binding.

In summary, binding of LZNV to YePEPT^K314A induces specific conformational changes, i.e., a rigid body movement of the N-terminal bundle towards the C-terminal bundle, and the partial formation of an intracellular gate (Y159). However, the distinct α-helix bending observed in inward-facing occluded peptide transporter structures is absent. Consequently, the LZNV bound structure appears to represent an intermediate state in the transition from the inward-open to the inward-facing occluded state. Finally, the PZ pocket is absent in the ligand-free state and forms upon binding of the LZNV inhibitor. Inspection of the available mammalian PEPT1 and PEPT2 structures[45,46] indicates that the PZ pocket is also absent in these LZNV-free structures (Supplementary Fig. 5) similar to our YePEPT^K314A apo structure (Fig. 6c). Thus and considering that LZNV is a potent inhibitor of PEPT1 and PEPT2, formation of the PZ pocket in these

mammalian peptide transporters upon LZNV binding would be expected.

**Binding of LZNV to humanised YePEPT variants**. LZNV is a potent inhibitor of human and mammalian PEPT1[52] and PEPT2[51]. As shown previously, binding of LZNV to YePEPT was enabled by the removal of the large, charged side chain of K314 (K314A mutant). Considering that the corresponding residue to K314 in PEPT1 and PEPT2, a glutamine (i.e., Q300 (PEPT1)/ Q319 (PEPT2), see Supplementary Table 2), has a relatively large side chain as well, leads to the question if LZNV binds to YePEPT if a glutamine is present at that position. To this aim, binding of LZNV to the humanised YePEPT^K314Q variant was evaluated using TSA (Fig. 5). YePEPT^K314Q did not show concentration-dependent stabilisation by LZNV. This result is consistent with the hypothesis that a large side chain in this position is blocking

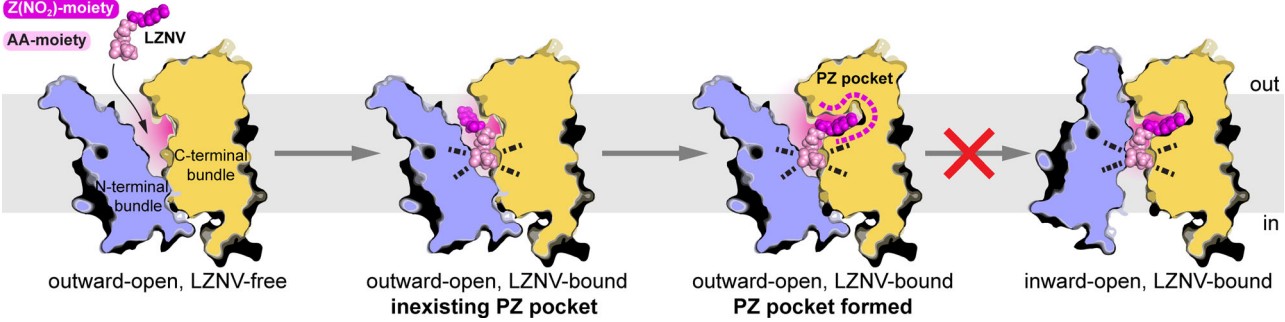

**Fig. 7 Proposed proton-dependent oligopeptide transporter (POT) inhibition mechanism by LZNV.** Cartoon of LZNV (dipeptide amino acid-moiety (AA-moiety): bright pink and $Z(NO_2)$-moiety: dark pink) approaching a peptide transporter in the outward-open state followed by its partial and full binding. Notice that the PZ pocket is initially absent and forms after binding of the conserved dipeptide backbone to the protein. The $Z(NO_2)$-moiety bound to the PZ pocket acts like a wedge impeding the transition to the inward-open state. LZNV-POT interactions are indicated by dotted lines. Whereas the grey bar represents the lipid bilayer, the N- and C-terminal bundles are coloured in blue and dark yellow, respectively.

the entry to the PZ pocket of YePEPT, thus abolishing binding of LZNV. However, this result is not consistent with the fact that mammalian PEPT1 and PEPT2 have a glutamine at the position corresponding to K314 in YePEPT and are efficiently inhibited by LZNV. Interestingly, the adjacent residue to K314 is a proline (P315) in YePEPT, while in PEPT1 and PEPT2 it is a glycine. The special and unique structural properties of proline as the only proteinogenic imino acid, impose strong restrictions on the conformation of the protein backbone. It is thus hypothesised that binding of LZNV to PEPT1 and PEPT2 is possible despite the large glutamine side chain, because of the larger conformational freedom due to the lack of the adjacent proline residue. To test this hypothesis, binding of LZNV was evaluated in the double humanised variant YePEPT^K314Q-P315G (Fig. 5). This variant shows a concentration-dependent stabilisation by LZNV similar to YePEPT^K314A supporting the hypothesis that access to the PZ pocket is generated by enhancing the conformational flexibility of the protein backbone around residue 314. In summary, this result strengthens the relevance of the observed binding mechanism of LZNV in YePEPT for the human and mammalian homologues PEPT1 and PEPT2.

**Proposed inhibition mechanism of proton-dependent oligopeptide transporters by LZNV.** It was shown that LZNV inhibits the transport of substrates competitively in PEPT1 and PEPT2 in different biological systems including mammalian cell lines (e.g., Caco-2 cells expressing human PEPT1 and SKPT cells expressing rat PEPT2) as well as when heterologously expressed in the yeast *Pichia pastoris* and in Xenopus oocytes[51,52]. Furthermore, it was demonstrated that LZNV can inhibit the uptake of [3H]Ala-Ala by the di- and tripeptide permease A (DtpA) in *E. coli* cells[54]. As we showed, the same is true for *E. coli* cells overexpressing YePEPT^K314A (Fig. 1c). For a similar inhibitor, Lys[Z(NO_2)]-Pro (LZNP), two-electrode voltage clamp experiments in oocytes showed that the uptake of Gly-Gln by human PEPT1 could also be inhibited when LZNP was administered intracellularly[50]. Oocyte experiments further demonstrated that LZNV and other similar modified dipeptides act as inhibitors and are not transported by PEPT1[52] and PEPT2[51].

Based on the here obtained and in the literature available information, we propose an inhibition mechanism of LZNV for POTs (Fig. 7). In a first step, the LZNV-moiety containing the dipeptide backbone binds to the POT interacting with conserved amino acid residues. After this partial binding of LZNV, the Z(NO2)-moiety induces formation of the initially absent PZ pocket through mainly rotamer changes of specific amino acid

side chains in the PZ pocket location. LZNV differs from substrates as it not only binds to both bundles in the conserved binding site, but also anchors its $Z(NO_2)$ moiety deep in the C-terminal bundle (i.e., in the PZ pocket). Thus, the $Z(NO_2)$-moiety bound to the PZ pocket might act like a wedge impeding the transition to the inward-open state.

## Conclusion

High-resolution structures of transport proteins with bound substrates or inhibitors are crucial for the understanding of their binding- and translocation mechanisms. Here we have presented the co-crystal structure of a potent PEPT1 and PEPT2 inhibitor (LZNV) bound to the POT-family member YePEPT^K314A at 2.66 Å resolution. In the binding pocket, six polar or ionic interactions between LZNV and amino acid side chains, one polar interaction with the backbone of YePEPT^K314A, and one π-stacking interaction with phenyl side chain of F318 were identified. Furthermore, three interactions between side chains of YePEPT^K314A, involved in the correct spatial orientation of amino acid side chains crucial for LZNV binding, were described. In thermal stabilisation studies, the influence of the identified side chains on overall binding of LZNV was evaluated. Importantly, a previously undescribed hydrophobic pocket accommodating the $Z(NO_2)$ moiety of LZNV was discovered in the C-terminal bundle of YePEPT^K314A. Comparison of the LZNV bound- and the apo structure of YePEPT^K314A allowed the identification of three aromatic residues, which undergo significant conformational changes upon inhibitor binding. This is accompanied by rigid body motion of about 5° of the N-terminal towards the C-terminal six-helix bundle. Interestingly, the PZ pocket is absent in the ligand-free state and forms upon binding of the LZNV inhibitor. Evaluation of LZNV binding in humanised versions of YePEPT demonstrated the importance of conformational flexibility of the protein backbone to enable access to the PZ pocket and strengthened the significance of the observed LZNV binding mechanism in YePEPT for human and mammalian PEPT1 and PEPT2. Based on these findings we proposed a possible inhibition mechanism of LZNV for proton-dependent oligopeptide transporters.

## Materials and methods

**Cloning**. The gene of wild-type YePEPT (UniPot accession number: A0A2R9TD79) from *Y. enterocolitica* was cloned into the pZUDF21 vector[60] as previously described[33]. The resulting construct pZUDF21-rbs-YePEPT-3C-His_{10} contains a C-terminal human rhinovirus 3C (HRV3C) cleavage site followed by a decahistidine-tag (His_{10}). Specific point mutations were introduced by site-directed mutagenesis using the QuikChange Lightning Multi Site-Directed Mutagenesis Kit (Agilent Technologies).

**Overexpression and membrane preparation**. In all, 24 l of Luria Bertani (LB) medium supplemented with ampicillin (100 μg/ml) were inoculated 1:100 from an overnight culture of *E. coli* BL21(DE3) pLysS transformed with pZUDF21-rbs-YePEPT-K314A-3C-His$_{10}$ and incubated at 37 °C and 180 rpm in an incubator shaker (Multitron, Infors HT). Heterologous protein overexpression was induced at an OD$_{600}$ of 0.6–0.7 by the addition of 300 μM isopropyl-β-D-thiogalactopyranoside (IPTG). 4 h after induction, cells were harvested by centrifugation (10,000 × *g*, 5 min, 4 °C) and the pellet was washed once with 2 l of membrane wash buffer (20 mM Tris-HCl, 500 mM NaCl, pH 8). After another centrifugation (10,000 × *g*, 5 min, 4 °C), the pellet was resuspended in 300 ml of lysis buffer (20 mM Tris-HCl, 50 mM NaCl, pH 8) and stored at −80 °C. For lysis, cells were thawed and sonicated for 60 min (total ON time) in 5 s ON/3 s OFF pulses using a tip sonifier (Branson 450 Digital Sonifier). Membranes were then pelleted by ultracentrifugation (150,000 × *g*, 1 h, 4 °C), washed once with 240 ml of membrane wash buffer and homogenised using a glass tissue homogeniser. The last ultracentrifugation was repeated once, the membranes resuspended in 36 ml of purification buffer (20 mM Tris-HCl, 300 mM NaCl, pH 8), flash-frozen in liquid nitrogen and stored at −80 °C.

**Purification of YePEPT$^{K314A}$**. YePEPT$^{K314A}$ membranes from 2 l of cell culture were solubilized in 7 ml of purification buffer supplemented with 2% (w/v) of n-undecyl-β-D-maltopyranoside (UDM, Glycon Biochemicals GMBH) for 1 h at 4 °C under gentle agitation. After ultracentrifugation (150,000 × *g*, 1 h, 4 °C), the supernatant was diluted with an equal volume of washing buffer (20 mM Tris-HCl, 300 mM NaCl, 5 mM L-histidine, 0.1% (w/v) UDM, pH 8), supplemented with 500 μl (bed-volume) Ni-NTA superflow resin (Qiagen) and incubated for 4 h at 4 °C under gentle agitation. Subsequently, the resin was transferred to a column (Promega Wizard Midicolumns) and washed with 15 ml of washing buffer and 3 ml of cleavage buffer (20 mM Tris-HCl, 150 mM NaCl, 0.075% (w/v) UDM, pH 8). The His-tag of YePEPT$^{K314A}$ was cleaved off by on-column digestion overnight at 4 °C using 470 μg/ml his-tagged human rhinovirus 3C protease[61] (HRV3C, BioVision, Milpitas, CA, USA) under gentle agitation in cleavage buffer. The cleaved protein was eluted by centrifugation (3000 × *g*, 1 min, 4 °C), residual HRV3C removed by another incubation with 100 μl (bed-volume) pre-equilibrated Ni-NTA superflow resin for 15 min at 4 °C under gentle agitation and followed by another centrifugation (3000 × *g*, 1 min, 4 °C).

**3D crystallisation**. YePEPT$^{K314A}$ crystals in the absence of LZNV were obtained as described previously[33]. For 3D crystallisation of YePEPT$^{K314A}$ with LZNV, the protein was concentrated to 7 mg/ml using a 50-kDa cut-off filter (Amicon Ultra 4, Merck Millipore). The ligand LZNV, which was synthesised as described previously[51], was directly dissolved in the concentrated protein solution at a final concentration of 20 mM (resulting in a molar ratio of ligand to protein of ~160). After 30 min of incubation on ice, the sample was ultracentrifuged (200,000×*g*, 10 min, 4 °C) before conducting crystallisation experiments. Crystallisation was performed using the sitting-drop vapour diffusion method. Drops were set up by mixing 150 nl protein and 150 nl crystallisation solution (50 mM glycine, 29–31% polyethylene glycol 300 (PEG300), pH 9.25) in 96-well polystyrene crystallisation plates (Swissci triple drop plates, Molecular Dimensions) using a Mosquito nanoliter crystallization robot (TTP Labtech). Crystals were fished after 14–30 days of incubation at 18 °C, flash-frozen and stored in liquid nitrogen until X-ray diffraction analysis.

**Structure determination and refinement**. All datasets of YePEPT$^{K314A}$ were collected at the X06SA (PXI) beamline of the Swiss Light Source (SLS; Paul Scherrer Institute, Villigen, Switzerland) using either a PILATUS 6 M (apo YePEPT$^{K314A}$) or an EIGER 16 M (LZNV bound YePEPT$^{K314A}$) detector (Dectris). In case of the LZNV bound YePEPT$^{K314A}$, two datasets from different crystals were indexed and integrated with XDS[62], then merged applying the CCP4 program suite[63] implemented BLEND program[64]. Except for the merging step, the processing procedure was the same for the apo YePEPT$^{K314A}$ dataset. To account for the anisotropic nature of the diffraction data, both datasets were further processed by the STARANISO software (http://staraniso.globalphasing.org/). The coordinates of the substrate-free inward-open conformation of YePEPT$^{WT}$ (PDB ID code 4W6V[33]) was used for structure solution by molecular replacement applying PHASER, a programme implemented in the Phenix software suite[65,66]. The final structures were obtained after iterative cycles of manual model building using COOT[67] and structure refinement runs applying phenix.refine including Translation/Libration/Screw (TLS) parameters (Phenix derived automatically assigned TLS groups were used[68]). All data collection, processing and refinement statistics can be found in Supplementary Table 1.

**Thermal shift assay using label-free differential scanning fluorometry**. The various YePEPT constructs were purified as described above, with the exception that the protein was not eluted by proteolytic cleavage of the His-tag, but by incubation in 20 mM Tris-HCl, 150 mM NaCl, 0.075% (w/v) UDM, 400 mM imidazole, pH 8 for 15 min at 4 °C under gentle agitation, followed by centrifugation (3000 × *g*, 1 min, 4 °C). The buffer of the eluted protein was then exchanged against TSA buffer (20 mM Bis-Tris propane-HCl, 150 mM NaCl, 0.075% (w/v) UDM, pH 8) using a desalting column (Zeba spin desalting columns 7k MWCO, Thermo Scientific). Samples for TSA measurements consisted of 4 μM of the respective YePEPT version and various concentrations of LZNV (i.e., 0 μM, 200 μM, 400 μM and 800 μM). Samples were prepared using an 8 mM LZNV stock solution in TSA buffer. Prior to the measurement, samples were incubated at room temperature for 30 min. The intrinsic fluorescence of tryptophan and tyrosine residues in the proteins were determined as the ratio of emission at 350 nm and 330 nm (F350/F330) with increasing temperatures from 35 °C to 95 °C and the inflation temperatures ($T_i$) calculated using the Tycho NT.6 device (NanoTemper Technologies GmbH). Differences in $T_i$ ($\Delta T_i$) for different ligand concentrations were calculated from the raw data by subtraction of the $T_i$ in the absence of LZNV using the Prism GraphPad 6 software.

**Uptake assays**. Uptake assays were performed as described previously[69]. In short, *E. coli* cells transformed with either YePEPT-K314A or empty vector were cultivated, resuspended in uptake buffer (50 mM HEPES-NaOH, 150 mM NaCl, 5 mM glucose, pH 7.5) at an OD$_{600}$ of 10 and stored on ice. Uptake was started by addition of resuspended cells (20 μl) to the reaction mixture at 18 °C (all in uptake buffer), i.e., 10 μl of 250 μM Ala-Ala spiked with [³H]Ala-Ala (Campro Scientific) to a specific activity of 0.02 Ci/mmol (5x substrate Master mix), and 20 μl of buffer or competitor (final reaction volume 50 μl). After 60 s, the reaction was stopped by addition of 450 μl ice-cold uptake buffer supplemented with 2.5 mM of Ala-Ala. For $IC_{50}$-determination, different LZNV concentrations (0.1 μM–7.5 mM) were applied. Samples were prepared using an 18.75 mM LZNV stock solution in uptake buffer. For $K_m$-determination, transport was measured at different Ala-Ala concentrations spiked with [³H]Ala-Ala to a specific activity of 0.0125 Ci/mmol. After reactions, washed cells (14,000 × *g*, 2 min, room temperature) were resuspended in 50 μl of 5% (w/v) sodium dodecyl sulfate, transferred to white 96-well plates (OptiPlate, PerkinElmer), 150 μl scintillation cocktail added (MicroScint 40, PerkinElmer) and incubated for 30 min at room temperature. Samples were then measured with a Packard TopCount scintillation counter (PerkinElmer). Raw data were corrected by subtraction of signals from the empty vector samples and normalised to the uninhibited signal. Nonlinear regression was used to determine $K_m$- and $IC_{50}$ values. All data were processed and plotted using the Prism GraphPad 6 software.

**Reporting summary**. Further information on research design is available in the Nature Research Reporting Summary linked to this article.

## Data availability
Relevant data are available from the corresponding author on reasonable request. Atomic coordinates for the apo- and LZNV bound crystal structures of YePEPT$^{K314A}$ have been deposited in the Protein Data Bank under accession numbers 7Q0L (apo) and 7Q0M (LZNV bound).

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

## Acknowledgements

We gratefully thank Kurt Hauenstein and Prof. Karl-Heinz Altmann (ETH Zurich) for providing LZNV, and the staff of the SLS (Paul Scherrer Institute) X06SA beamline for excellent support and advice. Financial support from the University of Bern, the Swiss National Science Foundation (SNSF; grant 310030_184980) and the NCCRs Molecular Systems Engineering and TransCure is kindly acknowledged.

## Author contributions

M.S., J.-M.J., H.I. and D.F. designed the experiments and analysed the data. Z.U. performed cloning and site-directed mutagenesis. M.S. overexpressed and purified YePEPT, co-crystalized YePEPT[K314A] and LZNV, and performed uptake experiments. H.I. performed TSA experiments. R.B. purified and crystalized YePEPT[K314A] for the apo structure. J.-M.J. solved and refined the crystal structures. M.S., J.-M.J. and D.F wrote the manuscript. All authors contributed to manuscript revision and approved the final version.

## Competing interests

The authors declare no competing interests.
