## [Peer Review File · Communications Chemistry]

Reviewers' comments:

Reviewer #1 (Remarks to the Author):

The manuscript by Stauffer et al. reports the first inhibitor-bound structure of a bacterial peptide transporter determined by X-ray crystallography. The work is based on an earlier reported crystal structure of the peptide transporter from *Yersinia enterocolitica* (YePEPT) and the mutant K314A which was shown to modulate substrate specificity. The authors could show binding and transport inhibition of YePEPT-K314A by the known PepT1 inhibitor LZNV, which is a modified Lys-Ala dipeptide. Therefore, they used thermal shift, radioactive whole-cell uptake, and inhibition assays to convincingly show the inhibitory function of LZNV. They then succeeded in crystallizing YePEPT-K314A in its apo and inhibitor bound form and unraveled the inhibitor coordination in the transporter binding site. While the “dipeptide”-moiety of the compound is similarly coordinated as in known POT-dipeptide structures (using the same set of conserved binding site residues), the authors discovered a new undescribed mostly hydrophobic pocket which they termed PZ pocket that hosts the modified lysine side chain of the inhibitor. The authors verified the interactions identified between the transporter and the inhibitor by mutational approaches and proposed an inhibition model for POTs by LZNV. The presented work represents a comprehensive and elaborate study. The experiments were well conducted and the results were nicely illustrated. The determined structures represent high-quality models and the entire study is highly valuable for the membrane transporter field.

I have some minor comments/questions the authors could address prior to publication:

- 1.) Can YePEPT wild type bind peptides, e.g. an unmodified KA dipeptide?
- 2.) Interactions of YePEPT-K314A with the dipeptide backbone of LZNV seem to follow the classical dipeptide binding mode – an additional supplementary figure showing an overlay of the new inhibitor bound structure and another known POT transporter structure bound to a dipeptide could confirm this and make it clear to the reader.
- 3.) Residue F318 seems to make a crucial interaction to the modified lysine side chain of the inhibitor. Did the authors attempt to mutate this residue and measure its binding affinity to LZNV?
- 4.) Lane 127: should read “than” instead of “that”
Lane 309 should read “Quikchange”
- 5.) What was the concentration of the stock solution of LZNV? Was the compound dissolved in DMSO, water, or buffer?
- 6.) Table S2: For completion – I recommend including the characterized and crystallized *E. coli* POTs DtpA and D

Reviewer #2 (Remarks to the Author):

The manuscript by Stauffer et al. reports the crystal structure of POT family peptide transporter from *Yersinia enterocolitica* (YePEPT1) in complex with a high affinity inhibitor LZNV, which is a modified dipeptide. YePEPT1 is a bacterial homologue of the mammalian peptide transporters PepT1 and PepT2, and previous studies by this lab and others have shown a strong mechanistic correlation between bacterial and eukaryotic POT family transporters. PepT1 and PepT2 are important targets for improving drug transport into the human body and physiologically play important roles in dietary peptide transport in the small intestine and kidneys.

This work presents important new insights into how to design high affinity inhibitors for peptide transporters. Specifically, the results show that inhibition by LZNV results from the peptide occupying a hydrophobic pocket PZ. (Previous studies had already established LZNV as a high affinity inhibitor of PepT1 and PepT2). The PZ pocket is new, joining previously described polar and hydrophobic pockets P1-3 from previous peptide bound structures. Similar to previous studies from other POT transporters, the PZ pocket forms in response to binding LZNV and appears to inhibit transport by simply jamming the helices in the C-terminal bundle.

Overall the paper is focused and well written, reporting novel and high quality data. I don't have any major concerns with this study. However, I did notice that in Fig.1 the concentrations of LZNV needed to stabilise the protein >200uM is two orders of magnitude higher than the IC50 of 7.6uM. Why the discrepancy? I wonder whether this might be due to the speed of temperature ramping in the tycho? It would be interesting to see if the concentration dependency of stabilisation is lower in the prometheus or rtPCR melting assay?

minor comments:

Fig.2a. residue numbers would be helpful here.

Fig.3 - a ligplot figure detailing a schematic of the ligand interactions might be helpful to readers.

Fig.6a - It might be clearer if the ribbons in the figure are bolded to highlight the structural changes.

The transporter also looks tilted in the membrane?

Supplementary figure S1 - please label the TMs. S3 - please label the side chains and helices.

Reviewer #3 (Remarks to the Author):

The study from Stauffer et al. presents the crystal structure of the peptide transporter YePepT-K134A from the bacteria *Yersinia enterocolitica* with and without the chemically modified dipeptide lysine[Z(NO₂)]-valine (LZNV), which is a known inhibitor of the human peptide transporters hPepT1 and hPepT2. The WT YePepT does not bind LZNV but the K134A mutant, which has a larger binding pocket relative to WT as discussed in a previous study (Boggavarapu et al. 2015) from the same laboratory. In this previous study, the WT YePepT crystal structure was determined and the dipeptide binding of YePepT WT and K134A was characterized.

Stauffer et al. compare the ligand binding site between the structures in complex with and without LZNV. The ligand is coordinated by three binding pockets, two of which are well characterized as pointed out in the study. The third pocket is accessible via the mutation K134A and is blocked in the WT. The third pocket is described and discussed appropriately in this study and is of interest to the peptide transporter field.

The structure analysis is well supported by state-of-the-art methods, such as thermostability analysis, radioactive ligand uptake and competition assay and point mutations. The paper is well-written and the figures are clear.

Overall, this is a relevant study for the field, contributing to the understanding of the promiscuity of the peptide transporters, that play an important role in nutrient uptake and drug delivery.

Here are a few minor comments:

- 1) The structure of WT YePepT was determined previously (Boggavarapu, 2015) and it should be included in the main text.
- 2) The inhibitor LZNV is relevant for hPepT1 and hPepT2. In this study, no binding of WT YePepT to LZNV is observed but only of the K314A mutant. The corresponding residue in hPepT1/hPepT2, Q300 and Q319 as stated in the Table S2, and their potential role on the newly characterized binding pocket (PZ pocket) in hPepT1 and hPepT2 context would be of interest to the reader in discussion.
- 3) In line 127, typo: "better that" -> "better than"
- 4) In line 127/128: it is stated that the LZNV inhibition of YePepTK314A is stronger than that of DtpA. Here, it would be valuable to include the IC50 value from the publication for the reader.
- 5) In line 182: I am not convinced that the residues in table S2 are similar and that one can expect a similar LZNV binding mode. Some residues are similar, some are not. Especially, the WT YePepT with K314 cannot bind LNzV whereas the mutant K314A can. In table S2, there are two further bacterial homologues with a corresponding lysine residue, which most likely do not bind LZNV.

Dr Yonca Ural-Blimke
Max Planck Institute of Biophysics, Frankfurt

Point-by-point response to Reviewers' comments for **Stauffer et al. 2021**
(Submission ID: COMMSCHEM-21-0326)

Reviewer #1

The manuscript by Stauffer et al. reports the first inhibitor-bound structure of a bacterial peptide transporter determined by X-ray crystallography. The work is based on an earlier reported crystal structure of the peptide transporter from *Yersinia enterocolitica* (YePEPT) and the mutant K314A which was shown to modulate substrate specificity. The authors could show binding and transport inhibition of YePEPT-K314A by the known PepT1 inhibitor LZNV, which is a modified Lys-Ala dipeptide. Therefore, they used thermal shift, radioactive whole-cell uptake, and inhibition assays to convincingly show the inhibitory function of LZNV. They then succeeded in crystallizing YePEPT-K314A in its apo and inhibitor bound form and unraveled the inhibitor coordination in the transporter binding site. While the “dipeptide”-moiety of the compound is similarly coordinated as in known POT-dipeptide structures (using the same set of conserved binding site residues), the authors discovered a new undescribed mostly hydrophobic pocket which they termed PZ pocket that hosts the modified lysine side chain of the inhibitor. The authors verified the interactions identified between the transporter and the inhibitor by mutational approaches and proposed an inhibition model for POTs by LZNV. The presented work represents a comprehensive and elaborate study. The experiments were well conducted and the results were nicely illustrated. The determined structures represent high-quality models and the entire study is highly valuable for the membrane transporter field.

Authors: We are grateful to the Reviewer for the positive feedback and for mentioning the relevance of our work for the membrane transporter field. The minor comments/questions raised by the Reviewer were addressed below.

I have some minor comments/questions the authors could address prior to publication:

1.) Can YePEPT wild type bind peptides, e.g. an unmodified KA dipeptide?

Authors: The specificity of YePEPT wild-type (YePEPT^{WT}) for different dipeptides was previously determined by Ala-Ala uptake inhibition – please see below panel c from Fig. 1 in Boggavarapu *et al.*, 2015, BMC Biology.

As seen from the panel, the mentioned KA dipeptide does not inhibit transport due to the preference of YePEPT^{WT} for acidic amino acid residues at the N-terminal position of dipeptides.

On the left: Panel c from Fig. 1 in Boggavarapu *et al.*, 2015, BMC Biology

2.) Interactions of YePEPT-K314A with the dipeptide backbone of LZNV seem to follow the classical dipeptide binding mode – an additional supplementary figure showing an overlay of the

new inhibitor bound structure and another known POT transporter structure bound to a dipeptide could confirm this and make it clear to the reader.

Authors: The authors thank the Reviewer for this suggestion that we have considered: please see the new Supplementary Figure 4 for an overlay of the YePEPT-K314A LZNV bound structure with a dipeptide-bound POT structure. For the latter, we chose the Ala-Phe bound structure of PepT_{st} (peptide transporter from *Streptococcus thermophilus*: PDB ID 4D2C; Lyons et al., 2014, EMBO Rep.).

3.) Residue F318 seems to make a crucial interaction to the modified lysine side chain of the inhibitor. Did the authors attempt to mutate this residue and measure its binding affinity to LZNV?

Authors: In an attempt to address this question, we expressed and purified YePEPT-K314A-F318A. However, the protein was severely destabilized by this mutation, as indicated by a reduction of the T_i by ~9 °C compared to YePEPT-K314A. Therefore, no further experiments were performed with the impaired YePEPT-K314A-F318A mutant.

4.) Lane 127: should read “than” instead of “that”

Lane 309 should read “Quikchange”

Authors: We thank the reviewer for noticing the mentioned misspellings. They are corrected in the revised manuscript.

5.) What was the concentration of the stock solution of LZNV? Was the compound dissolved in DMSO, water, or buffer?

Authors: For crystallization the LZNV powder was directly dissolved in the concentrated protein, leading to a final LZNV concentration of 20 mM. The concentrations of the initial stock solutions for uptake- and thermostability experiments were 18.75 mM and 8 mM, respectively. Each was prepared in the corresponding buffer.

6.) Table S2: For completion – I recommend including the characterized and crystallized E. coli POTs DtpA and D.

Authors: We agree with the Reviewer that it is valuable to add DtpA and DtpD in Table S2 (now called Supplementary Table 2). We therefore included both transporters in the revised version.

Reviewer #2

The manuscript by Stauffer et al. reports the crystal structure of POT family peptide transporter from *Yersinia enterocolitica* (YePEPT1) in complex with a high affinity inhibitor LZNV, which is a modified dipeptide. YePEPT1 is a bacterial homologue of the mammalian peptide transporters PepT1 and PepT2, and previous studies by this lab and others have shown a strong mechanistic correlation between bacterial and eukaryotic POT family transporters. PepT1 and PepT2 are important targets for improving drug transport into the human body and physiologically play important roles in dietary peptide transport in the small intestine and kidneys.

This work presents important new insights into how to design high affinity inhibitors for peptide transporters. Specifically, the results show that inhibition by LZNV results from the peptide occupying a hydrophobic pocket PZ. (Previous studies had already established LZNV as a high affinity inhibitor of PepT1 and PepT2). The PZ pocket is new, joining previously described polar and hydrophobic pockets P1-3 from previous peptide bound structures. Similar to previous studies from other POT transporters, the PZ pocket forms in response to binding LZNV and appears to inhibit transport by simply jamming the helices in the C-terminal bundle.

Overall the paper is focused and well written, reporting novel and high quality data. I don't have any major concerns with this study. However, I did notice that in Fig.1 the concentrations of LZNV needed to stabilise the protein >200uM is two orders of magnitude higher than the IC50 of 7.6uM. Why the discrepancy? I wonder whether this might be due to the speed of temperature ramping in the tycho? It would be interesting to see if the concentration dependency of stabilisation is lower in the prometheus or rtPCR melting assay?

Authors: We thank the Reviewer for the summary, which nicely reflects the performed work and for mentioning the relevance of our results for the research field. The authors are thankful for the positive feedback.

Concerning the question about the LZNV concentrations in the thermostabilization experiments:

We expect that the higher concentrations of LZNV needed are attributed to the fact that thermostability experiments are conducted using purified detergent-solubilized protein, while uptake experiments are performed with cells where the target transporter is embedded in a lipid membrane. Similar behaviors were also observed with other transporters in our laboratory.

minor comments:

Fig.2a. residue numbers would be helpful here.

Authors: The authors thank the Reviewer for this suggestion that will improve the clarity of Fig. 2a. We have added the amino acid sequence of the displayed transmembrane helix in the corresponding figure legend and labeled the first and last amino acid in panel A of Fig. 2a.

Fig.3 - a ligplot figure detailing a schematic of the ligand interactions might be helpful to readers.

Authors: We agree that a 2D depiction of the protein-ligand interactions in form of a LigPlot would be helpful for readers. Therefore, we included it as Supplementary Figure 3.

Fig.6a - It might be clearer if the ribbons in the figure are bolded to highlight the structural changes. The transporter also looks tilted in the membrane?

Authors: Thank you for this suggestion. Accordingly, we created and introduced into the revised manuscript an alternative version of the panel (Fig. 6a) with bolder helices to make the structural changes more visible. We restricted the bolding to the concerned region, in order to keep displayed key residues and ligand nicely visible.

Supplementary figure S1 - please label the TMs.

Authors: Done – please see new Supplementary Figure 1.

S3 - please label the side chains and helices.

Authors: Done – please see new Supplementary Figure 5 (Suppl. Fig. 3).

Reviewer #3

The study from Stauffer et al. presents the crystal structure of the peptide transporter YePepT-K134A from the bacteria *Yersinia enterocolitica* with and without the chemically modified dipeptide lysine[Z(NO₂)]-valine (LZNV), which is a known inhibitor of the human peptide transporters hPepT1 and hPepT2. The WT YePepT does not bind LZNV but the K134A mutant, which has a larger binding pocket relative to WT as discussed in a previous study (Boggavarapu et al. 2015) from the same laboratory. In this previous study, the WT YePepT crystal structure was determined and the dipeptide binding of YePepT WT and K134A was characterized.

Stauffer et al. compare the ligand binding site between the structures in complex with and without LZNV. The ligand is coordinated by three binding pockets, two of which are well characterized as pointed out in the study. The third pocket is accessible via the mutation K134A and is blocked in the WT. The third pocket is described and discussed appropriately in this study and is of interest to the peptide transporter field.

The structure analysis is well supported by state-of-the-art methods, such as thermostability analysis, radioactive ligand uptake and competition assay and point mutations. The paper is well-written and the figures are clear.

Overall, this is a relevant study for the field, contributing to the understanding of the promiscuity of the peptide transporters, that play an important role in nutrient uptake and drug delivery.

Authors: The authors thank the Reviewer for the encouraging and positive feedback, and for pointing out the quality and relevance of our work for the field.

Here are a few minor comments:

1) The structure of WT YePepT was determined previously (Boggavarapu, 2015) and it should be included in the main text.

Authors: We thank the Reviewer for this comment. In the revised version, we have now included this information in the first sentence of the last paragraph of page 4.

2) The inhibitor LZNV is relevant for hPepT1 and hPepT2. In this study, no binding of WT YePepT to LZNV is observed but only of the K314A mutant. The corresponding residue in hPepT1/hPepT2, Q300 and Q319 as stated in the Table S2, and their potential role on the newly characterized binding pocket (PZ pocket) in hPepT1 and hPepT2 context would be of interest to the reader in discussion.

Authors: As mentioned and discussed in the manuscript, removal of the large, charged K314 residue is essential for binding of LZNV to the ligand binding site of YePEPT. To address the question from the Reviewer, we mutated K314 to Q314 (K314Q), i.e., we introduced the same residue as found in hPepT1 and hPepT2 into YePEPT. As displayed in the updated version of Fig. 5, K314Q does not bind LZNV (i.e., no significant thermostabilization is observed).

Interestingly, the residue adjacent to K314 in YePEPT is a proline, which is known to restrict conformational flexibility, while in hPepT1 and hPepT2 the residue adjacent to Q300/Q319 is a glycine, which allows for high flexibility. This led to the hypothesis that binding of LZNV to the human transporters is enabled by a higher local flexibility at the entry to the PZ pocket. We then tested this hypothesis by introducing higher flexibility into YePEPT^{K314Q} by exchanging P315 with a glycine. In fact, the resulting (double humanized) YePEPT^{K314Q-P315G} variant showed similar thermal stabilization as YePEPT^{K314A}, which strongly supports the flexibility hypothesis. We have added these new results (Fig. 5) and its discussion in the revised version of the manuscript, and thank the Reviewer for this suggestion that has further strengthened our manuscript.

3) In line 127, typo: “better that” -> “better than”

Authors: We thank the Reviewer for noticing this typo. It was corrected in the revised manuscript.

4) In line 127/128: it is stated that the LZNV inhibition of YePepTK314A is stronger than that of DtpA. Here, it would be valuable to include the IC50 value from the publication for the reader.

Authors: We agree with the Reviewer and have added the value in the revised manuscript.

5) In line 182: I am not convinced that the residues in table S2 are similar and that one can expect a similar LZNV binding mode. Some residues are similar, some are not. Especially, the WT YePepT with K314 cannot bind LNzV whereas the mutant K314A can. In table S2, there are two further bacterial homologues with a corresponding lysine residue, which most likely do not bind LZNV.

Authors: A significant number of residues forming the PZ pocket are comparable, sometimes even identical, e.g., F386^{YePEPT} is identical in all listed organisms except in 3 from 13 where the corresponding residue is a tryptophan or tyrosine (still being aromatic as F386^{YePEPT}). Another example is Q313^{YePEPT}: this amino acid is fully conserved in all 13 organisms. However and indeed, there are also less well conserved residues. Therefore, we have attenuated the statement in this sentence, e.g., by replacing “similar” by “comparable”. The point about wild-type YePEPT (K314), which can be extrapolated to other bacterial peptide transporters with a lysine at that position has been addressed and clarified previously in point 2), and by new, additional experiments (including discussion) in the revised version of the manuscript.

Dr Yonca Ural-Blimke

Max Planck Institute of Biophysics, Frankfurt

REVIEWERS' COMMENTS:

Reviewer #1 (Remarks to the Author):

I am happy with the way that my comments and suggestions have been addressed in the revised version of the manuscript. I recommend publishing the manuscript in its current form.

Reviewer #2 (Remarks to the Author):

The authors have addressed my comments and I'm happy to support publication of the revised Ms. good job!

Reviewer #3 (Remarks to the Author):

Thank you for addressing all the points. The new results with the (double) humanized YePepT are especially interesting.